# Isolation and Grading of Faults in Battery Packs Based on Machine Learning Methods

**Sen Yang** [1,2], **Boran Xu** [1,2] and **Hanlin Peng** [2,*]

1    System Engineering Research Institute of China State Shipbuilding Co., Ltd., Beijing 100036, China;
     wesen@ncepu.edu.cn (S.Y.); 201811000814@ncepu.edu.cn (B.X.)
2    Department of Mechanical Engineering, North China Electric Power University, Baoding 071003, China
*    Correspondence: hlp@ncepu.edu.cn

**Abstract:** As the installed energy storage stations increase year by year, the safety of energy storage batteries has attracted the attention of industry and academia. In this work, an intelligent fault diagnosis scheme for series-connected battery packs based on wavelet characteristics of battery voltage correlations is designed. First, the cross-cell voltages of multiple cells are preprocessed using an improved recursive Pearson correlation coefficient to capture the abnormal electrical signals. Secondly, the wavelet packet decomposition is applied to the coefficient series to obtain fault-related features from wavelet sub-bands, and the most representative characteristic principal components are extracted. Finally, the artificial neural network (ANN) and multi-classification relevance vector machine (mRVM) are employed to classify and evaluate fault mode and fault degree, respectively. Physical injection of external and internal short circuits, thermal damage, and loose connection failure is carried out to collect real fault data for model training and method validation. Experimental results show that the proposed method can effectively detect and locate different faults using the extracted fault features; mRVM is better than ANN in thermal fault diagnosis, while the overall diagnosis performance of ANN is better than mRVM. The success rates of fault isolation are 82% and 81%, and the success rates of fault grading are 98% and 90%, by ANN and mRVM, respectively.

**Keywords:** battery fault diagnosis; recursive correlation coefficient; artificial neural network; relevance vector machine

## 1. Introduction

Li-ion batteries are extensively used in electric vehicles (EVs) and their safety has aroused wide concerns [1]. Most of the spontaneous combustion incidents of EVs are confirmed to be caused by battery failures [2], which can be induced by mechanical damage [3], electrical overload [4], and thermal abuse [5]. For the safe operation of EVs, it is critical to accurately identify the fault state of battery packs. In response, diverse fault diagnosis and control techniques were reported to improve the safety of battery systems [6].

Model-based diagnostic methods estimate battery state of health by establishing a physical characteristic model or identifying the residuals between measured and model parameters. To describe the thermal runaway process, Ouyang et al. [7] proposed an energy transfer image method to quantify the reaction of battery materials, which makes the chain reaction mechanism of thermal runaway and internal short-circuit fault clearer. In [8], Dey et al. detected and evaluated thermal failure levels based on a one-dimensional temperature field model and a partial differential equation observer. Hashemi et al. [9] estimated and modeled the parameters of the battery with the machine learning technique to achieve accurate fault diagnosis. These methods rely too much on the model's accuracy, and most of the models are affected by noise, interference, and unmolded characteristics. Although physical coupling models have improved reliability and accuracy, they consume mass simulation and computing resources and thus are only suitable for online real-time

applications [10]. Seo et al. [11] proposed a model-based switching method to estimate the abnormality of cell resistance for internal short-circuit detection.

Knowledge-based diagnostic methods try to understand fault mechanisms and rely on long-term accumulated knowledge and experience. Xiong et al. in [12] proposed a rule-based detection method for over-discharge detection using established temperature voltage rules whereby a failure warning is directly given by a Boolean expression. Muddappa et al. [13] incorporated voltage, temperature and SOC residuals into fuzzy rules to detect various fault types, including over-charge/discharging and abnormal aging. Huber et al. in [14] introduce optical inspection means for the classification of battery separator defects, and expert knowledge and machine learning were used in the diagnosis process. Though this method can achieve high accuracy and robustness, the involved dedicated instruments are expensive.

Data-driven diagnostic methods try to obtain potential fault features and patterns by directly analyzing system-running data without the requirement of accurate analytical models or the understating of complicated fault mechanisms. Chen et al. [15] realized battery failure detection by evaluating the local deviation of observed data using a local outlier factor based on the Grubbs criterion. In [16], Yao et al. simulated the battery charging and discharging process in a vibration environment to observe voltage fluctuations and then used the entropy transfer to realize the detection of connection faults. Hong et al. [17] proposed a thermal runaway prediction scheme based on the big data and information entropy, and the location of thermal faults in the battery pack can be accurately located. Machine learning techniques were also extensively explored in battery fault diagnosis benefitting from the competent capability in nonlinear characteristic approximating and automatic decision making [18]. Yao et al. [19] exploited the grid search support vector machine using features extracted by a modified signal covariance matrix, whereby battery fault states can be identified timely and efficiently. Xie et al. [20] downsized the dimensions of randomly selected features of time-domain statistics with principal component analysis and the refined features are fed into a relevance vector machine to make diagnostic decisions on battery faults. To achieve sensitive battery anomaly detection, Schmid et al. [21] devised a robust studentized outlier sample method to select the principle feature components derived from cross-cell monitoring data. Ojo et al. [22] proposed an approach relying on the long short-term memory neural network, in conjunction with an alteration to the walk-forward technique, to accurately estimate battery surface temperature for thermal fault diagnosis. Yang et al. [23] used the artificial neural network to estimate short-circuit current and then predicted the maximum temperature increase as well as internal and surface temperature distribution of the faulty cell based on a 3D electro-thermal coupling model. In [24], Xue et al. determined the diagnostic coefficient based on the statistical distribution of the operational data from a cloud monitoring platform, and three screening methods were designed to detect and locate battery faults. Hong et al. [25] utilized the classical deep learning algorithm, i.e., long short-term memory recurrent neural network, to accurately predict battery voltage, which provided data support for battery fault diagnosis.

Though acceptable results on battery fault diagnosis were reported, there are still many problems regarding the fault diagnosis of large-scale battery packs. For example, the inconsistency inside battery packs makes accurate characteristic modeling tough work, and thus the robustness of model-based diagnostic methods is unsatisfactory; classical data-driven methods suffer from the random load dynamics which can make decision-making strategies often give false alarms. This paper proposes an intelligent model-free diagnostic scheme: the cross-level voltages inside the pack are monitored and recursively correlated to filter load dynamics, and then artificial neural networks (ANN) and multi-classification relevance vector machine (mRVM) are employed to realize adaptive fault distinguish and evaluation. The main innovative contributions are listed as follows:

- Physical fault injection experiments on battery packs to collect a realistic fault dataset;

- Enhance fault location capability through cross-cell voltage, and improved recursive Pearson correlation (RPC) to shield fault-irrelevant dynamics such as measurement noise and load fluctuations;
- Extract and refine fault features from wavelet sub-bands of RPC sequences, with which ANN- and mRVM-based fault diagnosis frameworks are constructed.

## 2. Sensor Topology and Signal Preprocessing

### 2.1. Cross-Level Voltage

As shown in Figure 1, cross-cell sensors, i.e., V1~Vn, measure the sum potentials of two neighbor batteries, whereby various fault signals can be covered effectively.

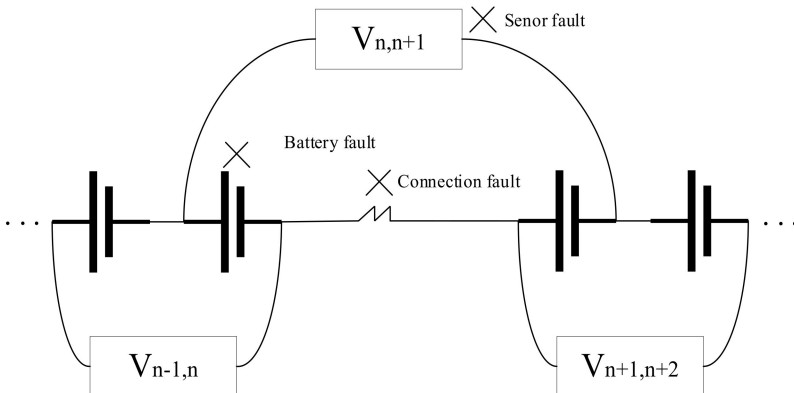

**Figure 1.** Connection of battery pack and sensors.

### 2.2. Recursive Pearson Correlation (RPC)

The classical Pearson correlation cannot work in real-time for online correlation analysis and is less sensitive to short-term anomalies. Therefore, recursive and forgetting mechanisms are introduced to obtain the correlation between two sequences of $X = (x_1, x_2, \cdots)$ and $Y = (y_1, y_2, \cdots)$ as:

$$r(X, Y)_i = \frac{wP_k - Q_k R_k}{\sqrt{wS_k - Q_k{}^2}\sqrt{wT_k - R_k{}^2}} \tag{1}$$

$$\begin{cases} x_i = V_1 + \varphi_i + N_i, y_i = V_2 + \varphi_i + M_i \\ P_i = P_{i-1} + x_i y_i - x_{i-w} y_{i-w} \\ Q_i = Q_{i-1} + x_i - x_{i-w} \\ R_i = R_{i-1} + y_i - y_{i-w} \\ S_i = S_{i-1} + x_i{}^2 - x_{i-w}{}^2 \\ T_i = T_{i-1} + y_i{}^2 - y_{i-w}{}^2 \end{cases} \tag{2}$$

where $i$ is time epoch, $\omega$ is window size, $x$ and $y$ are cross-cell voltages, $\varphi$ is a square wave to eliminate oscillations caused by minor noises or interferences during system static state, and $N$ and $M$ are Gaussian white noises. $w$ is used to maintain a proper length of data to avoid the short-term features being overwhelmed and to accommodate certain dynamics. By introducing the small square signal $\varphi$, the oscillation of $r(x, y)_i$ caused by measurement error in a steady-state can be alleviated. The height and width of $\varphi$ needs to be adjusted with respect to the real-time load. On this basis, the improved correlation coefficient can effectively alleviate false alarms caused by load fluctuations while retaining critical information.

### 3. Fault Diagnosis Methodology

*3.1. Method Overview*

As shown in Figure 2, this paper presents a generic fault diagnostic scheme. The RPC time series of neighbor voltages are disassembled into multiple sub-bands by wavelet transform, and fourteen indexes are gathered from the bottomed sub-bands to reflect system state information. PCA is used to refine these indexes to principal components (PC) which are fed to ANN and mRVM as inputs for model training and resultant models can be applied for online fault diagnosis. As for fault degree, the severity of faults is classified into four levels: no fault (healthy), minor, moderate, and critical.

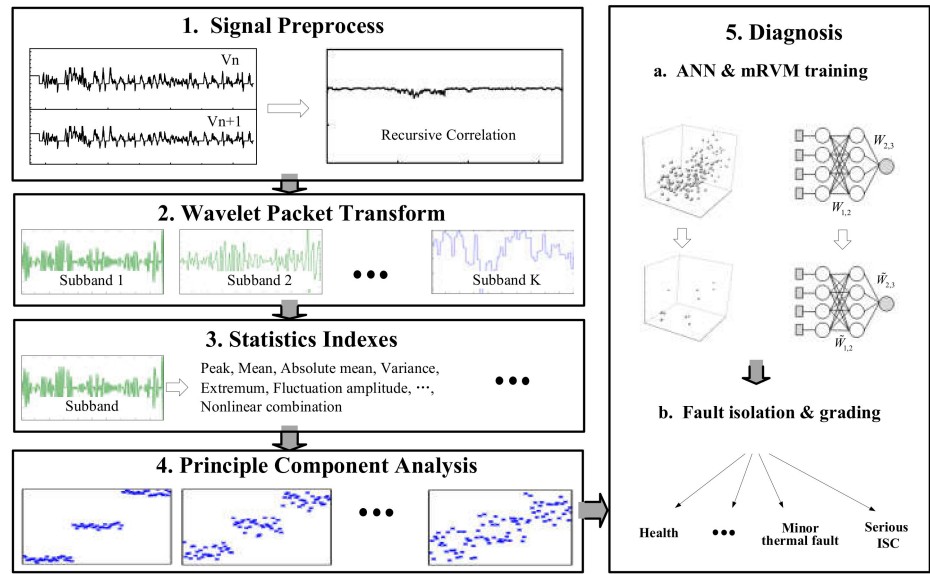

**Figure 2.** Schematic diagram of the proposed diagnosis framework.

*3.2. Wavelet Packet Transform (WPT)*

Fourier transform cannot give signal time-domain information; the multi-resolution mechanism of wavelet analysis can decompose time-frequency effectively. WPT derives a family of wavelets from the primary basis function:

$$\Psi(j,k) = \frac{1}{2^{j/2}}\Psi\left(\frac{t - 2^j k}{2^j}\right) \tag{3}$$

where $2^j$ and $2^j$ are responsible for the scaling and moving in the time domain, respectively. For $x(t)$, the decompositions are:

$$C(j,k) = \frac{1}{2^{j/2}}\int x(t)\Psi^*\left(\frac{t - 2^j k}{2^j}\right) dt \tag{4}$$

where $C(j,k)$ is the basis function. From the top layer signal, each parent layer can be divided into two orthogonality sequences as:

$$\begin{cases} H(z) = h_0 + h_1 z^{-1} + \cdots + h_{L-1} z^{-(L-1)} \\ G(z) = g_0 + g_1 z^{-1} + \cdots + g_{L-1} z^{-(L-1)} \end{cases} \tag{5}$$

Thus, two neighbor sequences can be obtained by the following recursion:

$$\begin{cases} c_{l+1}^{2p}(n) = \sqrt{2}\sum_k h(k) c_l^p(2n - k) \\ c_{l+1}^{2p+1}(n) = \sqrt{2}\sum_k g(k) c_l^p(2n - k) \end{cases} \tag{6}$$

where $N$ is the length of the discrete sequence, and $n = 0, 1, 2, \cdots, N - 1$.

### 3.3. Principle Component Analysis (PCA)

As shown in Table 1, in each DWPT sub-band of the RPC sequence, fourteen statistical indexes are designed to characterize fault characteristics. Since not all these indexes are helpful for fault diagnosis, PCA is employed to extract the most representative principal components. Given the feature matrix $X = \{ x_{n,m} | m = 1, 2, \ldots, M, n = 1, 2, \ldots, N \}$, each column is normalized as:

$$\widetilde{x}_m = \frac{x - \text{mean}(x)}{\sqrt{\text{var}(x)}} \tag{7}$$

where $\text{var}(x)$ means variance. Then the covariance matrix is:

$$C = \frac{1}{M} \widetilde{X} \widetilde{X}^T \tag{8}$$

**Table 1.** Statistics of WPT sub-bands.

| Index | Specification | Index | Specification |
|---|---|---|---|
| $p_1$ | $\max|x|$ | $p_8$ | $p_1 / p_5$ |
| $p_2$ | $\sum_{n=1}^{N} x(n) / N$ | $p_9$ | $p_1 / p_4$ |
| $p_3$ | $\sum_{n=1}^{N} |x(n)| / N$ | $p_{10}$ | $\sum_{n=1}^{N} (x(n) - p_2)^2 / N$ |
| $p_4$ | $\left( \sum_{n=1}^{N} \sqrt{|x(n)|} / N \right)^2$ | $p_{11}$ | $\sum_{n=1}^{N} \left( \frac{x(n) - p_2}{\sqrt{p_{10}}} \right)^3 / N$ |
| $p_5$ | $\sqrt{\sum_{n=1}^{N} x(n)^2 / N}$ | $p_{12}$ | $\sum_{n=1}^{N} \left( \frac{x(n) - p_2}{\sqrt{p_{10}}} \right)^4 / N - 3$ |
| $p_6$ | $p_5 / p_3$ | $p_{13}$ | $\max|x(n)| - \min|x(n)|$ |
| $p_7$ | $p_1 / p_3$ | $p_{14}$ | $\sum_{n=1}^{N} (x(n) - p_2)^3 / N$ |

Arrange the eigenvectors of $C$ according to their eigenvalues and the first k rows form a transformation matrix $P$. The original matrix $X$ can be compressed into the k-dimensional matrix:

$$Y = PX \tag{9}$$

In addition, a cumulated contribution $a_p$ can be defined as:

$$a_p = \frac{\sum_{i=1}^{k} \lambda_i}{\sum_{k=1}^{M} \lambda_i} \tag{10}$$

where $\lambda_i$ is the eigenvalue. After this, the dimension of the original matrix can be significantly reduced, which is beneficial to the training and testing of the subsequent diagnosis model.

### 3.4. ANN-Based Diagnostic Model

Neurons are the core functional elements of the ANN and are organized in a particular connection mode. Take the $i$th neuron as an example to exemplify the relationship between data vector $X = [x_1, x_2, \ldots, x_n]^T$ and weight vector $W = [w_1, w_2, \ldots, w_n]^T$. The input–output relation of a neuron $i$ is:

$$net_i = \sum_{j=1}^{n} w_{ij} x_j - k \tag{11}$$

$$y_i = f(net_i) \tag{12}$$

where $x_j$ is the input, $w_{ij}$ is the connection weight from neuron $j$ to $i$, $k$ represents a bias, $f$ is the activation transfer function, and *net* is the net activation. In order to improve the classification performance, a multi-layer ANN model is designed. The first active function is linear:

$$f_l(x) = kx + c \tag{13}$$

The second active function is the s-type function:

$$f_s(x) = \frac{1}{1 + e^{-ax}}(0 < f_s(x) < 1) \tag{14}$$

where $f_s(x)$ can map the input of neurons to the interval (0, 1). The training rule of adaptive learning based on gradient descent makes the generalization ability of the network better and reduces the difficulty of determining the optimal network structure. In this paper, two ANN models are instantiated to determine fault type and evaluate fault level, respectively.

*3.5. mRVM-Based Diagnostic Model*

mRVM simplifies decision-making by removing uncorrelated points using automated correlation decisions to obtain sparse models. mRVM extends RVM to multi-classification problems by introducing auxiliary regression targets and weight parameters. Moreover, probabilistic likelihoods are given as the confidence of classification given dataset $X = \{x_i, t_i\}(i = 1, 2, \ldots, N)$, where $t$ is the label for each sample category, $K = (k_1, k_2, \ldots, k_n), K \in R^{N*N}$, $k_n$ is the similarity between the input sample $n$ and other samples. An auxiliary variable $Y \in R^{L*N}$ is used as the mRVM regression objective, and model parameters $w \in R^{N*L}$ are introduced as the weight parameter, which follows the normal distribution of $(0, a_{nl}^{-1})$. $A \in R^{N*L}$ belongs to the scale matrix, then the noise model can be obtained as:

$$y_{\ln}|\omega_l, k_n \sim N_{y_{\ln}}\left(\omega_l^T k_n, 1\right) \tag{15}$$

Convert regression targets in (15) to label categories:

$$t_n = i, y_{ni} > y_{nj}, \forall i \neq j \tag{16}$$

The probability output of class members is:

$$P(t = i|\omega, K_n) = \varepsilon_{p(u)}\left\{\prod_{j \neq i}\Phi\left(u + (\omega_i - \omega_j)^T k_n\right)\right\} \tag{17}$$

where $\mu$ follows the standard Gaussian distribution, $\Phi$ is the Gaussian cumulate distribution, and the probability of $\omega$ is given as:

$$P(\omega|Y) \propto P(Y|\omega)P(\omega|A) \propto \prod_{l=1}^{L} N\left(\left(KK^T + A_l\right)^{-1}Ky_l^T, \left(KK^T + A_l\right)^{-1}\right) \tag{18}$$

According to (18), the MAP estimator is:

$$\hat{\omega} = \text{argmax}_W P(W|Y, A, K) \tag{19}$$

The maximum posterior estimation weight is updated as:

$$\hat{\omega}_l = \left(KK^T + A_l\right)^{-1}Ky_l^T \tag{20}$$

The posterior probability distribution of prior parameters of the weight vector $\omega$ is:

$$P(A|\omega) \propto P(W|A)P(A|p, q) \propto \prod_{l=1}^{L}\prod_{n=1}^{N} G\left(\frac{p+1}{2}, \frac{\omega_{nl}^2 + 2q}{2}\right) \tag{21}$$

Then, features and the corresponding labels are fed to train the model. Here in this work, the classic type-II maximum likelihood method [26] is employed to train the model by incrementally adding samples to the model and finally some relevance vectors are reserved as the skeleton of the sparse model based on a contribution criterion. Then, when new features appear, the probability of each candidate's result can be given. In this paper, two mRVM models are instantiated for fault type isolation and fault level evaluation, respectively.

## 4. Experimental Setup

As Figure 3 depicts, in order to obtain real fault data, a platform is built up with two battery test cabinets, a battery pack with four Li-ion cells (NCM, 3.7 V, 4 Ah) in series, a fault-injection module and a host computer. The load current is the Federal Urban Driving Schedule (FUDS). Herein, one test cabinet (60 V/50 A, 0.1% FS) is responsible for pack charging and discharging, and the other test cabinet (5 V/200 A, 0.02% FS) has four channels which are used to measure the potentials of the four cells. Four kinds of faults including internal short-circuit (ISC), external short circuit (ESC), poor cell connection (PCC) and thermal damage (THD) are simulated with the configuration in Table 2. The fault-injection module controls a motor and several relays to simulate PCC fault and ESC fault: PCC fault is simulated by intermittently connecting the inter-cell resistor which is passively driven by the vibration from the motor; ESC is simulated by contacting cell electrodes with different ESC resistors by the relays. ISC is induced by electric abuse of overcharge and THD is simulated with a heat gun in which jet heat flows on the cell surface.

With the abuses in Table 2, different failures can be induced. A total of 600 data samples including four failure modes (PCC, ISC, THD, ESC) and no-fault mode are measured for fault type classification (FTC). Moreover, fault degree is defined in four levels: healthy, minor fault, moderate fault, and critical fault, and 600 data samples are also collected for fault degree evaluation (FDE).

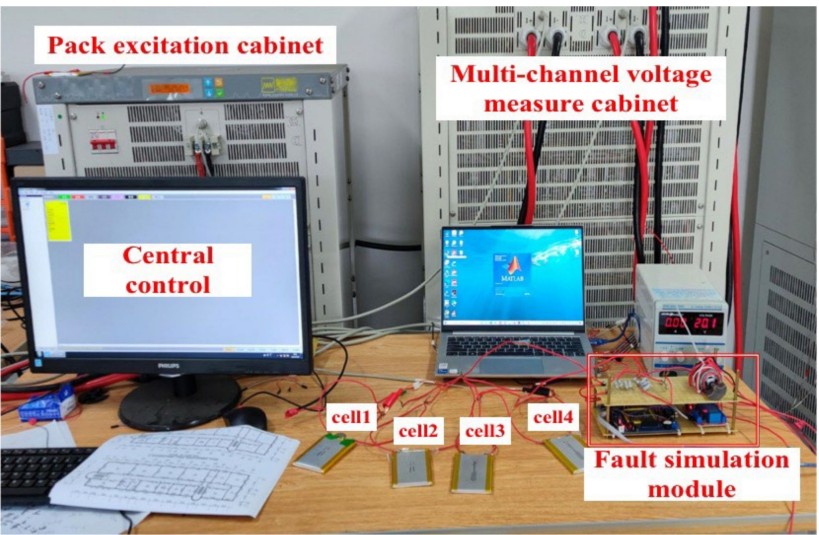

**Figure 3.** Physical view of experimental setup.

**Table 2.** Abuse configuration to induce different faults.

| Type | Degree | Abuse of Operational Details | |
|---|---|---|---|
| ISC | | Number of overcharges | Overcharge capacity (%) |
| | I | 4 | 130 |
| | II | 7 | 140 |
| | III | 10 | 130 |
| ESC | | ESC resistance (mΩ) | Duration (s) |
| | I | 20 | 0.1, 0.3, 0.6 |
| | II | 50 | 0.1, 0.3, 0.6 |
| | III | 100 | 0.1, 0.3, 0.6 |
| PCC | | Inter-cell resistance (Ω) | Motor voltage (V) |
| | I | 20 | 10~15 |
| | II | 20 | 16~24 |
| | III | 20 | 25~31 |
| THD | | Temperature (°C) | Heating time (min) |
| | I | 300 | 3 |
| | II | 400 | 4 |
| | III | 450 | 5 |

## 5. Experimental Verification

### 5.1. Feature Extraction

According to experiment data, the RPC curves of different faults may have similar characteristics, therefore, it is difficult to obtain direct evidence in the original RPC sequences for fault isolation and evaluation. Therefore, the WPT is introduced to analyze the sequences. More abundant details can be obtained as decomposition goes deep. By checking signal decomposition performance concerning the sensitivity to fault anomaly with different configurations, Daubechies' basis function is selected to analyze the sequences with five-layer decomposition at three lengths of 50, 100, and 300, respectively. Among the 32 sub-bands, four sub-bands with the most efficacious features are retained. From the indexes in Table 1, a vector containing $3 \times 4 \times 14 = 168$ elements can be obtained. Here, the PCA is used to reduce the dimension of the vectors. When the number of remained PCs increases to 18, the cumulative weight reaches 94.2%. That is, the information included in the raw features can be effectively refined into fewer components.

Here, mRVM training tests are carried out to evaluate the deduction of PC number. As the remained PCs increase from 1 to 13, the average accuracy increases from 60.9% to 96.1%, and the average remained relevance vectors increases from 87 to 183, i.e., 11.1% and 23.3% of the samples, respectively. When 3 PCs remain, the accuracy rate is 91.0%, from which, as the PCs increase from 3 to 13, the accuracy rate only increases slightly by 5.1% while the relevance vectors increase twice. Although more input features can improve classification performance, the sparsity of the model is impaired, resulting in high computational complexity and poor generalization. In order to balance model overfitting and classification accuracy, 3 PCs are retained for the subsequent training and testing.

### 5.2. Fault Diagnosis Based on ANN

#### 5.2.1. Fault Isolation

A multi-layer ANN is constructed for fault isolation and trained with the adaptive learning rule of gradient descent. The first hidden layer has ten neurons using the linear activation function, and the second hidden layer is configured with three neurons using the sigmoid activation function. Different training samples, iteration times, target errors, and training speeds in the ANN system can cause differences in recognition performance. As shown in Table 3, the optimal model with the maximum number of iterations of 2000 and the learning rate of 0.01 are selected. Then, three vector sets containing 500 FTC samples are fed into the model for training.

**Table 3.** Training recognition rate of the model using different parameters.

| Iterations Learning Rate | 500 | 1000 | 1500 | 2000 | 2500 |
|---|---|---|---|---|---|
| 0.005 | 77% | 85% | 82% | 83% | 83% |
| 0.01 | 83% | 83% | 82% | 86% | 82% |
| 0.02 | 83% | 84% | 82% | 83% | 83% |
| 0.05 | 84% | 85% | 82% | 84% | 83% |

Another 100 FTC samples of data are used as test samples. Figure 4 shows the classification performance of the five fault states in different colors: (circular, connection fault), (sphere, ESC fault), (square, ISC fault), (starriness, thermal fault) and (triangle, no fault). The samples with distinct colors are the misjudged ones. As can be seen, only a few samples are misjudged. In Figure 4a, there are six "PCC" samples that are misjudged as other faults. In Figure 4b, only one "ESC" sample is misjudged as "PCC". In Figure 4c, two "ISC" samples are misjudged as "No fault". In Figure 4d, most "THD" samples are misjudged as other faults and the success rate is only 33.67%. In Figure 4e, only one sample is misjudged. In summary, thermal faults are the most difficult to identify, and the success rate of fault isolation for all cases is 82%.

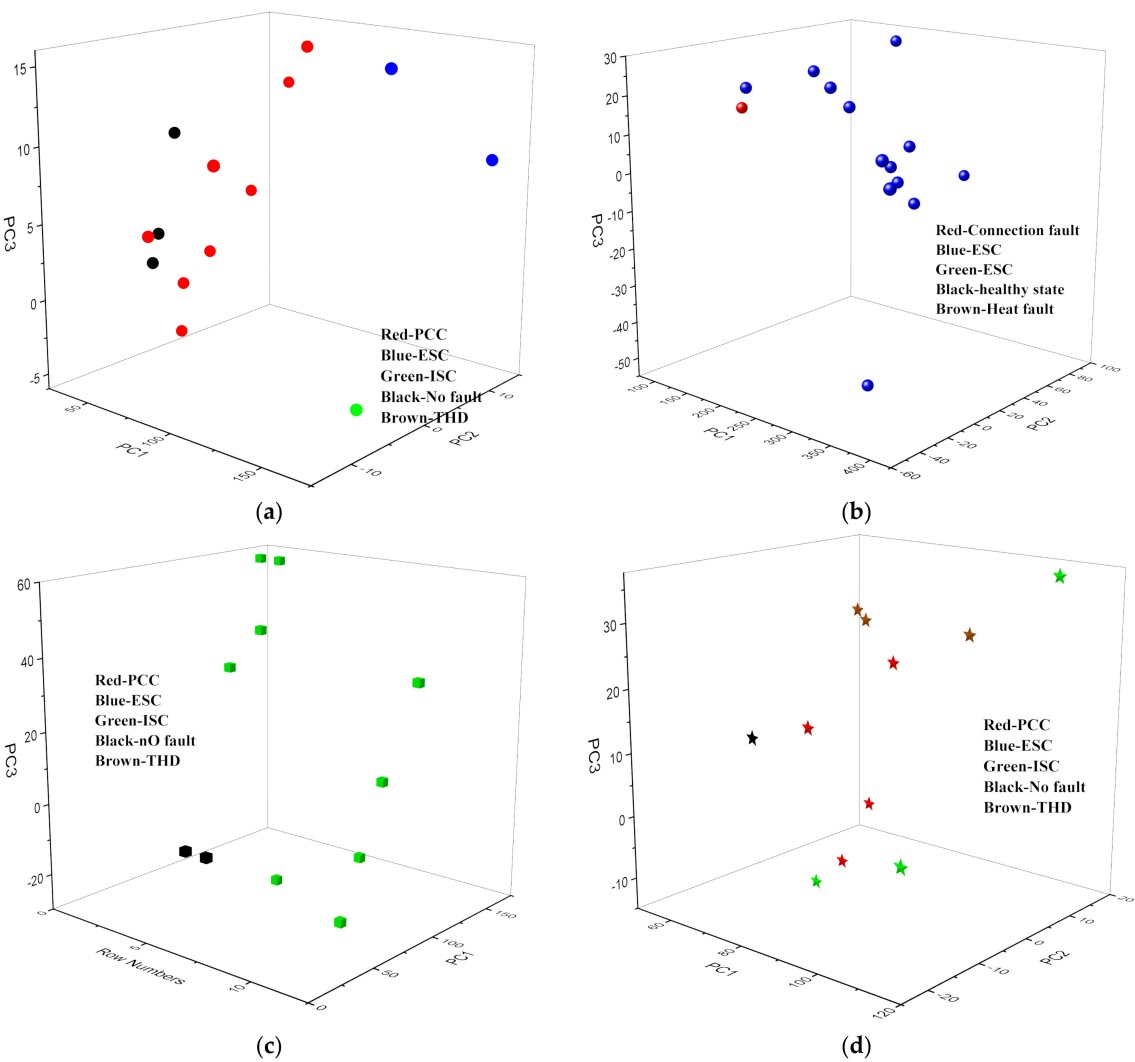

**Figure 4.** *Cont.*

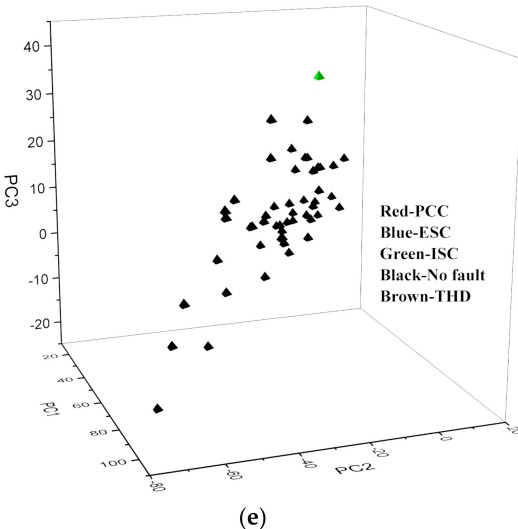

(**e**)

**Figure 4.** Fault isolation performance based on ANN: (**a**) PCC; (**b**) ESC; (**c**) ISC; (**d**) THD; (**e**) No fault.

### 5.2.2. Fault Grading

A total of 500 FDE samples are used to train the ANN. Then, another 100 FDE samples are fed into the resultant model to give grading results. In Figure 5, the marks with different colors have different meanings: round-grade IV fault (critical), spherical-grade III fault (moderate), square-grade II fault (minor), triangle-grade I no fault (healthy). Figure 5a shows that the distributions of different fault degrees are relatively independent and the overall success rate is relatively high. The results in Figure 5b–e show that only a few data are misjudged. In conclusion, the overall success rate of identifying the severity of the fault is 98%.

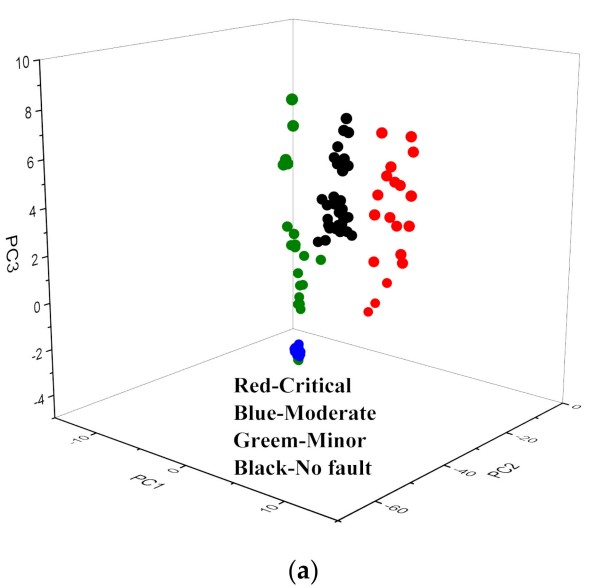

(**a**)

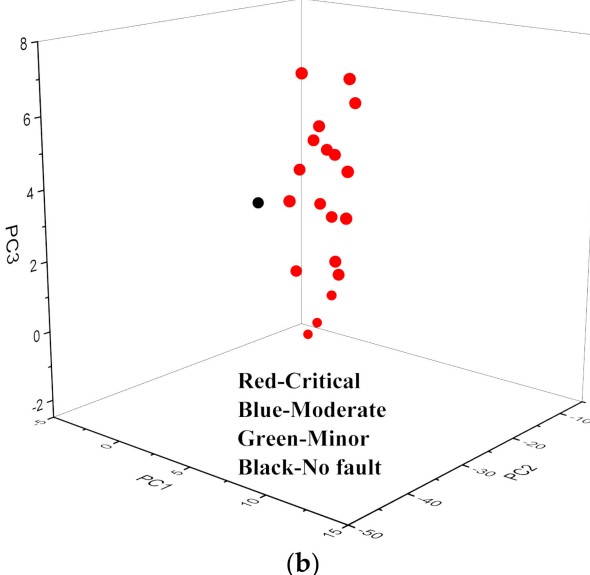

(**b**)

**Figure 5.** *Cont.*

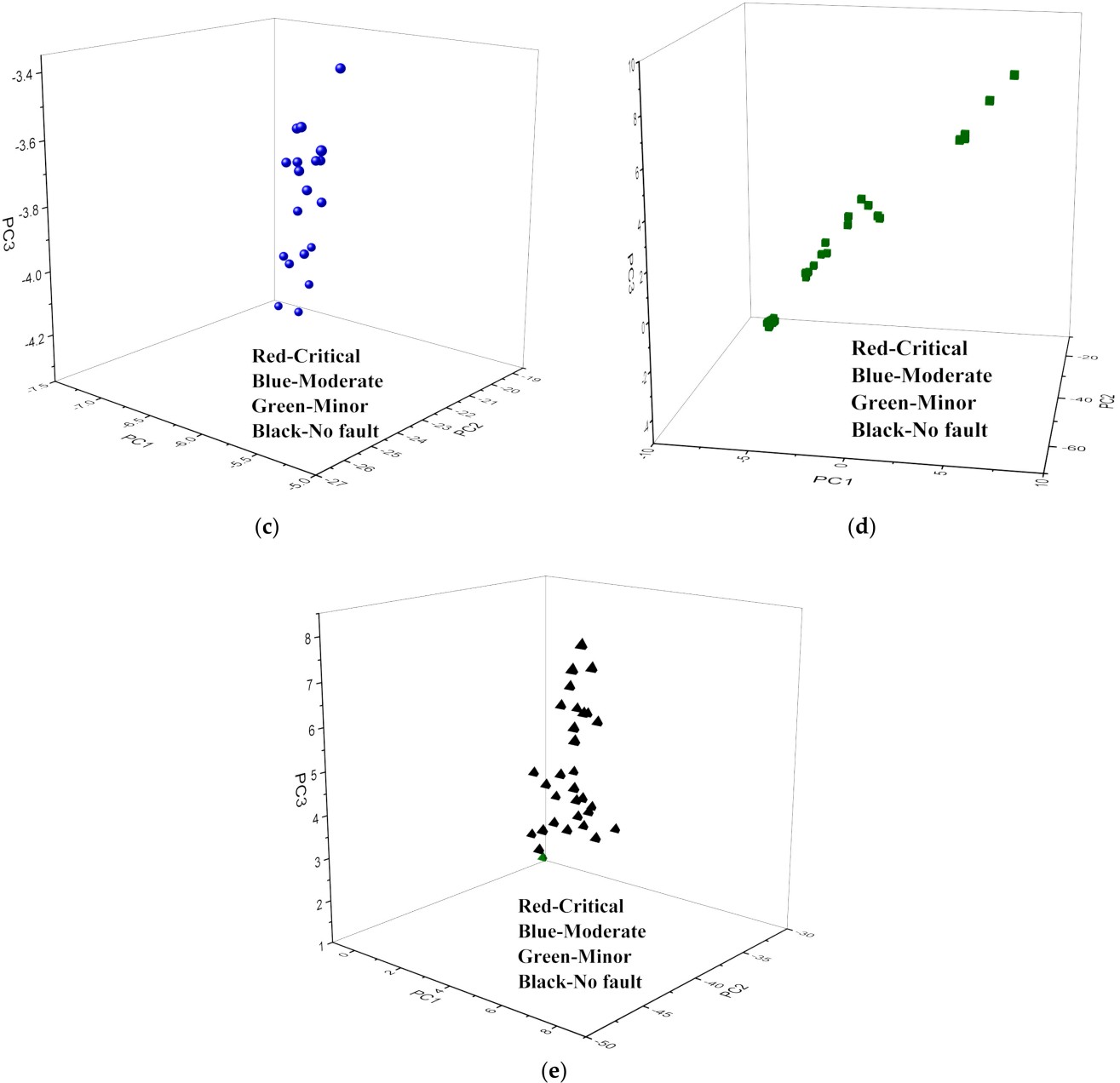

**Figure 5.** Fault grading performance based on ANN: (**a**) Overview; (**b**) Critical fault; (**c**) Moderate fault; (**d**) Minor fault; (**e**) Healthy.

*5.3. Fault Diagnosis Based on mRVM*

5.3.1. Fault Isolation

In order to deal with the nonlinearity of feature space, the Gaussian function is used as the kernel function of mRVM in this work. The involved parameters are determined based on the gradient descending rule. Each fault type is given a posteriori probability as confidence, indicating the degree to which it belongs. The maximum confidence is determined as the classification result. Figure 6a–e show the fault classification results on different FTC data samples, and different shape markers represent different faults. There are two small sphere colors in Figure 6a, which mean erroneous judgments. Figure 6b shows some "ESC" samples are misjudged as "PCC", while in Figure 6c, a small number of "ISC" samples are judged as "No fault" or "PCC". In Figure 6d, several "THD" samples are determined as other fault types, with a successful isolation rate of 35%. In Figure 6e, only

one sample is misjudged. To sum up, thermal faults are the most difficult to identify, and the overall success rate of fault isolation is 81%.

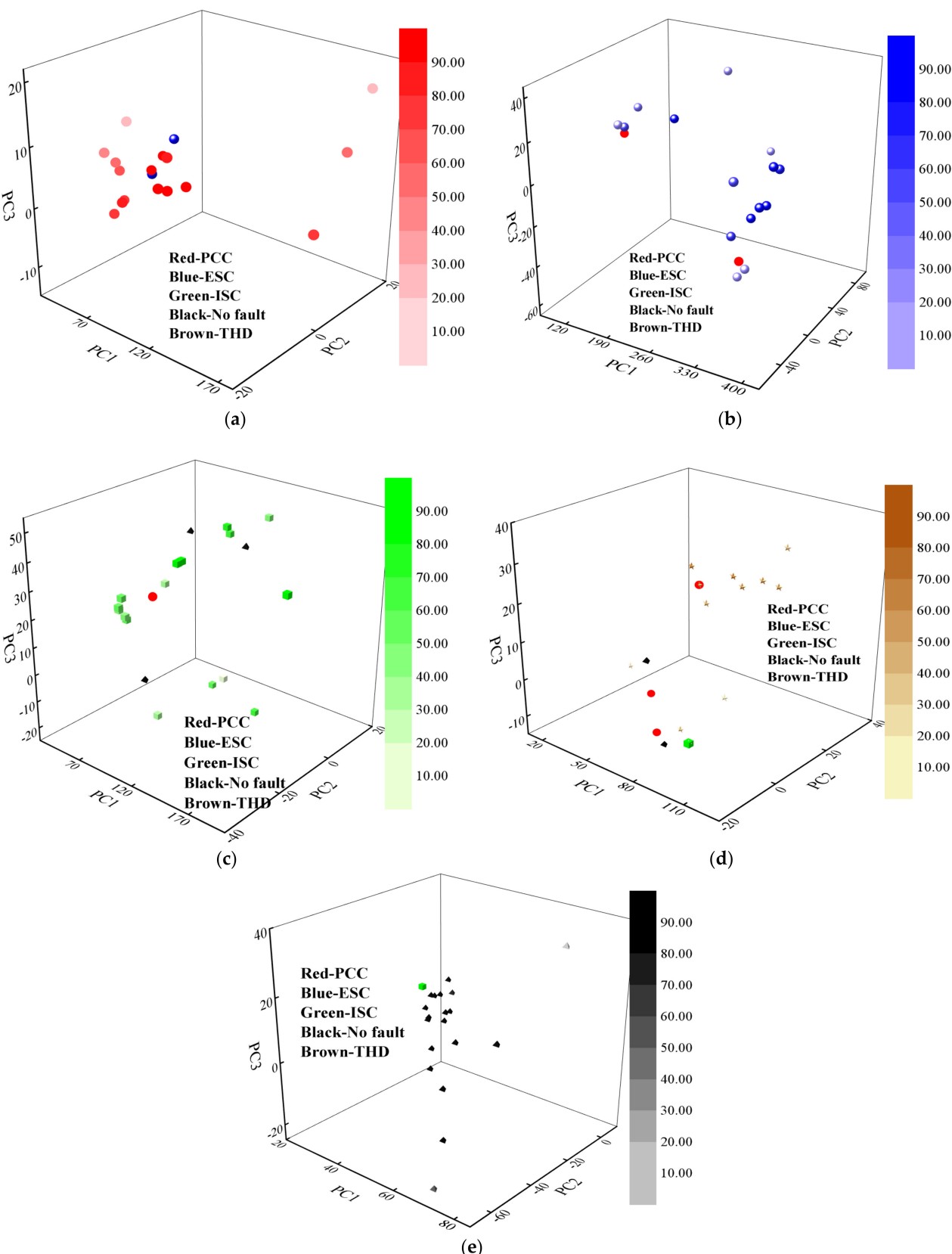

**Figure 6.** Fault isolation performance based on mRVM: (**a**) PCC; (**b**) ESC; (**c**) ISC; (**d**) THD; (**e**) No fault.

### 5.3.2. Fault Grading Based on mRVM

Similarly, 500 and 100 FDE samples are used to train and test the mRVM model, respectively. Figure 7a–e show the classification results, in which the shapes with distinct colors are misjudged ones. Most types of misjudged faults are prone to be underestimated regarding severity. For example, three of the "critical fault" samples in Figure 7b are judged to be "No fault", and four of the "moderate fault" samples in Figure 7c are judged to be "minor fault". In Figure 7d, a few "minor fault" samples are judged to be "No fault". Nevertheless, some faults are overestimated, such as in Figure 7e, one "health" sample is judged as "minor fault". In conclusion, the success rate of identifying the severity of all cases is 90%.

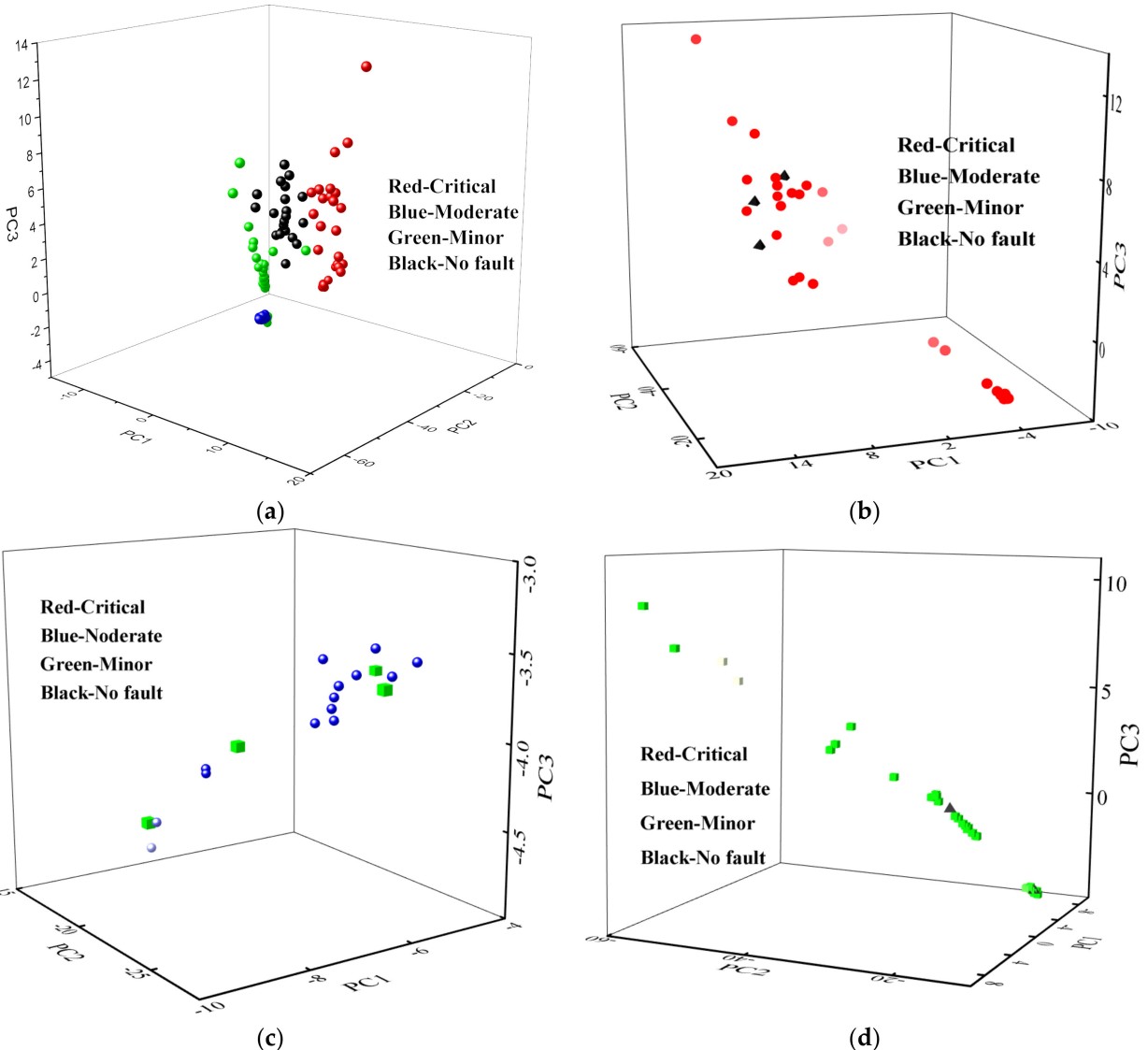

**Figure 7.** *Cont.*

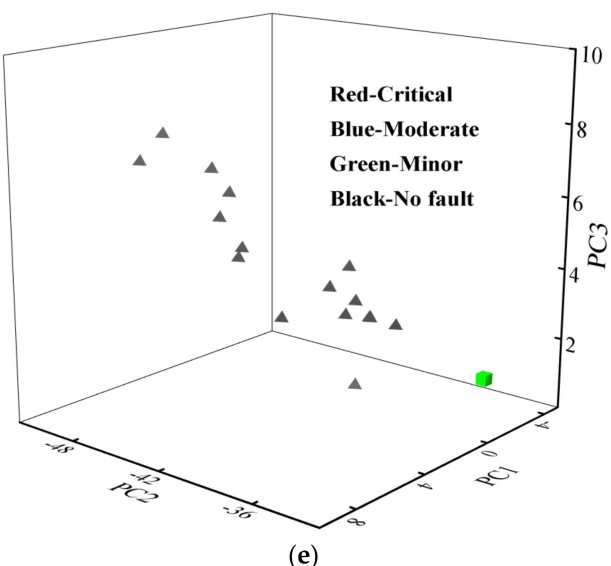

(**e**)

**Figure 7.** Fault grading performance based on mRVM: (**a**) Overview; (**b**) Critical fault; (**c**) Moderate fault; (**d**) Minor fault; (**e**) Healthy.

Table 4 lists the performance statistics of all the above fault isolation and grading experimental results by items and in total. In summary, ANN is superior to mRVM though mRVM is more consistent in different isolation cases, and most notably in fault grading. mRVM is outclassed by ANN in all cases but it is worth noting that both models cannot deal with THD isolation very well because different intensities of thermal abuse may cause similar damage to the cell and the anomalous features tend to be clustered.

**Table 4.** Statistics of diagnosis performance (success rate %) with the ANN and mRVM models.

| | Isolation | | | Grading | |
|---|---|---|---|---|---|
| | **ANN** | **mRVM** | | **ANN** | **mRVM** |
| PCC | 57% | 90% | Critical | 95% | 91% |
| ESC | 93% | 90% | Moderate | 100% | 78% |
| ISC | 82% | 80% | Minor | 100% | 94% |
| THD | 64% | 70% | No fault | 100% | 94% |
| No fault | 98% | 95% | | | |
| Total | 82% | 81% | | 98% | 90% |

## 6. Conclusions

This paper presents two online diagnosis schemes for common faults in battery packs based on machine learning techniques. Neighbor cell voltages in a pack are correlated with the improved Pearson correlation coefficient whereby system electrical anomalies can be sensed and load fluctuation and noise can be effectively eliminated. The wavelet packet transform is then used to perform time-frequency decomposition on the correlation sequences. The characteristics of decomposed sub-bands are obtained and refined as key principle features by PCA-based dimension-reduction. Then the ANN and mRVM models are employed to use the extracted features for fault diagnosis. The experimental results show that the proposed methods have good performance in fault detection, classification, and evaluation. For mRVM, the success rate of fault isolation is 81%, and the success rate of fault grading is 90%. For ANN, the success rate of fault isolation and grading is 82% and 98%, respectively. Although the overall diagnosis performance of ANN is superior to mRVM in most cases, mRVM gives better results regarding the most intractable thermal fault identification. In our future work, we will further study the recognition of thermal

fault patterns and utilize more advanced classification models to achieve more robust diagnosis performance.

**Author Contributions:** Conceptualization, S.Y. and H.P.; methodology, S.Y. and H.P.; validation, S.Y., B.X. and H.P.; formal analysis, B.X.; investigation, S.Y.; resources, H.P.; data curation, B.X.; writing—original draft preparation, S.Y. and B.X.; writing—review and editing, H.P.; visualization, B.X.; supervision, H.P. All authors have read and agreed to the published version of the manuscript.

**Funding:** This research received no external funding.

**Data Availability Statement:** All the details of this work, including data and algorithm codes, are available by contacting the corresponding author: hlp@ncepu.edu.cn.

**Conflicts of Interest:** The authors declare no conflict of interest.

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
