# Peer review of "Isolation and Grading of Faults in Battery Packs Based on Machine Learning Methods"

_electronics, doi:10.3390/electronics11091494_

Round 1

Reviewer 1 Report

This work proposed a method for faults classification and grading in battery packs based on machine learning models. Topic is interesting, however it will be requested to solicit extensive English editting to improve the quality of writing and better deliver the meaning of this work.

In the Introduction section, it is claimed that diverse fault diagnosis methods have been proposed. Please elaborate it with more details and related literature. Moreover, please provide a consolidated taxonomy of such methods.

In line 57, 'Machine learning is competent in data potential information analysis'. It is not appropriate. Please rephrase it.

In terms of the contributions, the first one is related to the data collection process. It should be better described rather than a simple description of action of 'trigger'. Meanwhile, it will be appreciated to share the data set publicly. 

Figure 2 needs substaintial improvement to provide sufficient information for the framework.

What is the meaning of 'double-layer'?

While dataset and the method including feature extraction appear to be valid, it will be expected to explain why and how these machine learning models are selected as there are many other models available off the shelf. 

Data sample is small. Please clarify how the hyperparameters are determined.

Any comparison with the state-of-the-art literature methods? If not, how is the model evaluated for practical usage?

Reviewer 2 Report

 In this paper authors have devised a fault diagnosis framework for series-connected battery packs based on wavelet characteristics of battery multi-level voltage correlations.

For authors of this paper, I have the following concerns:

  1. Usually, a system used for error diagnosis purpose has a very careful design about the front-end sensors. However, the physical implementation and parameter detail are not evident in this manuscript. Authors have provided only a little parametric explanation about system specifications.
  2. The introduction section should be improved by clarifying the similarities and differences between the related work and the proposed method. In current form the contribution of the paper seems very marginal. Authors should emphasize and clearly describe their contribution. It is recommended that authors divide introduction in subsections like i). Background, ii). Similar works, iii). Research gap, iv). Contribution.

  3. The parameters of used techniques such as recursive correlation, wavelet transform, and PCA should be clearly described and justified.
  4. Specify the implementation details/parameters of all machine learning algorithms and the used parameetrs tuning approach. Also justify the choice of used algorithms.
  5. It is recommended to add a separate section “Dataset” with a clear description.
  6. The presented results to be well discussed and analyzed.
  7. It is recommended to separate results and discussion sections. Properly support conclusions in the discussion section by providing findings and figures/values. What's the influence of the parameters in the proposed method? I suggest author to give the discussion of the main parameters. Also include a comparison with state-of-the-art concurrent solutions in the discussion section, preferable in a Tabular form.

Round 2

Reviewer 1 Report

It has resolved my concerns.